

# A polyphasic approach in the identification and biochemical characterization of *Dunaliella tertiolecta* with biodiesel potential from a saltern in Mauritius

Kamlesh Ramdhony[1], Daneshwar Puchooa[1], Turki Kh. Faraj[2], Abdulwahed Fahad Alrefaei[3], JunFu Li[4] and Rajesh Jeewon[4,5]

[1] Department of Agricultural and Food Science, Faculty of Agriculture, University of Mauritius, Réduit, Mauritius
[2] Department of Soil Science, College of Food and Agriculture Sciences, King Saud University, Riyadh, Saudi Arabia
[3] Department of Zoology, College of Science, King Saud University, Riyadh, Saudi Arabia
[4] Kunming Institute of Botany, Chinese Science Academy, Kunming, Yunnan, China
[5] Department of Health Sciences, Faculty of Medicine and Health Sciences, University of Mauritius, Réduit, Mauritius

Corresponding author
Rajesh Jeewon, r.jeewon@uom.ac.mu

## ABSTRACT

Bioprospecting robust and oleaginous strain is crucial for the commercialization of microalgae-based biodiesel. In this study, a microalgal strain SCH18 was isolated from a solar saltern located in Mauritius. This isolate was identified as *Dunaliella tertiolecta* based on a polyphasic approach that combined molecular, physiological, and morphological analyses. Furthermore, the effect of different salinities on the biochemical composition and fatty acid profile of this microalga was investigated to explore its potential in producing biodiesel. Results from the growth studies showed that salinity of 1.0 M NaCl was optimal for achieving a high growth rate. Under this salt concentration, the growth rate and the doubling time were calculated as $0.39 \pm 0.003$ day$^{-1}$ and $1.79 \pm 0.01$ days, respectively. In terms of biochemical composition, a substantial amount of carbohydrate ($42.02 \pm 5.20\%$), moderate amount of protein ($30.35 \pm 0.18\%$) and a low lipid content ($17.81 \pm 2.4\%$) were obtained under optimal NaCl concentration. The fatty acid analysis indicated the presence of palmitic acid, stearic acid, palmitoleic acid, oleic acid, linoleic acid, gamma, and alpha-linolenic acids, which are suitable for biodiesel synthesis. The predicted biodiesel properties were in accordance with the standard of ASTM 6751, indicating that the microalgal isolate *D. tertiolecta* SCH18 is a potential candidate for use in biodiesel production.

## INTRODUCTION

Interest in the production of energy from renewable resources has increased in recent years. This has been mainly motivated by growing concern about the future availability of fossil fuels and greenhouse gas emissions, which is recognized as the main contributor to

global warming. Biodiesel is a clean and sustainable alternative to petroleum diesel. This biofuel offers several advantages over diesel in terms of biodegradability, superior lubricity, higher flash point, and lower toxicity (*Enwereuzoh, Harding & Low, 2020*). Currently, the main feedstock for biodiesel production includes oils derived from oilseed crops such as soybeans, sunflower, rapeseed, and palm. Since most of these crops are also used in human and animal nutrition, policymakers and researchers have raised concerns about food insecurity resulting from an increased demand for biofuels. It is argued that large-scale oilseed crops cultivation could raise food prices and weaken food production by competing over agricultural resources such as arable land, freshwater and chemical fertilizers. For this reason, there is a growing focus on using microalgae as an alternative to oilseed crops.

Microalgae have attracted renewed interest from researchers because of their ability to accumulate considerable amounts of lipids that can be converted into biodiesel. Moreover, compared to terrestrial crops, microalgae are not limited by season and can grow throughout the year with a short growth cycle. Most importantly, they do not compromise food production, as they can be grown on marginal land using water sources that are not suitable for crops or human consumption (*Ishika, Moheimani & Bahri, 2017*). Beyond biofuel production, microalgae have remarkable ability to fix $CO_2$ from flue gases (1.88 kg $CO_2$ per 1 kg of biomass) and remediate wastewater by assimilating nutrients (nitrogen and phosphorus) (*Llamas et al., 2021*; *Valdovinos-García et al., 2022*). In this sense, microalgae cultivation can help reduce environmental pollutants, all while producing biomass which can be converted into energy and valuable products. However, to commercialize microalgae-based biodiesel, it is imperative to develop low-cost and large biomass production technologies. In this regard, open-pond cultivation of microalgae using readily available and inexpensive resources such as seawater is highly desirable. Open ponds are one of the simplest biomass production systems in terms of construction and operation. However, such systems are highly susceptible to contamination by undesirable predators and grazers, which can lead to low biomass productivity or cause crash of the entire culture. To mitigate this problem, some researchers have suggested the cultivation of microalgae under high saline conditions as a strategy to reduce biological contamination by less salt-tolerant species (*Park et al., 2018*). In this regard, the search for an oil-producing strain that can resist high salt levels seems particularly relevant. However, it should be noted that high salinity-induced osmotic stress can adversely affect the physiology and metabolic behavior of microalgae. According to *Khatoon et al. (2014)*, exposure to high salt concentration induces alterations in the growth and biochemical composition of cultured microalgae, even if they are tolerant to high salinity. Therefore, investigating the effect of salinity stress on microalgal growth and biochemical content is crucial for finding the best operating conditions to maximize biofuel production.

Solar salterns, also known as salt pans, are a potential source for the bioprospecting of salt-tolerant strains of microalgae. This artificial structure of interconnected ponds is generally subjected to extreme salinity conditions, with the NaCl concentration increasing from that of seawater to levels greater than 5.5 M (*Gammoudi et al., 2022*). Microalgae such as those of the genus *Dunaliella* are well known for their exceptional ability to survive

in high salinity environments. Their survival in such hostile setting is due to their adaptation mechanisms which include the activation of $Na^+$ and $K^+$ ion membrane transporters, modification of membrane permeability and fluidity, up-regulation of key enzymes involved in energy metabolism, *de novo* synthesis of osmoprotectant and antioxidant molecules (*Polle et al., 2020*). Among the different antioxidant molecules synthesized by *Dunaliella*, β-carotene which accumulates up to 14% dry weight of cell, has found extensive use in the pharmaceutical, nutraceutical and animal nutrition industries due to its strong free radical scavenging ability and coloring properties (*Barbosa et al., 2023*). In addition to β-carotene, the biomass of this microalga has been found to contain other valuable compounds such as lipids, proteins, carbohydrates and polyunsaturated fatty acids (*Gharajeh et al., 2020*). These metabolites have shown application potential for various industrial purposes, from improving human and animal health through the formulation of nutritional supplements and feed, to production of biofuels (*da Silva et al., 2021*). By virtue of these multifaceted attributes, *Dunaliella* has gained prominence in biotechnology and industry.

Currently, there are 28 recognized species in the *Dunaliella* genus; five have been documented from freshwater environments and the remaining 23 species from saline and hypersaline water bodies such as salt lakes and salterns (*Pourkarimi et al., 2020*) Over the past few years, several representatives of this genus have been isolated from various hypersaline sources worldwide, including Sambhar Lake, India (*Singh et al., 2019b*), Sidi Ameur salt lake, Algeria (*Yaiche-Achour et al., 2018*), Salinas de Manaure, Colombia (*Gallego-Cartagena, Castillo-Ramírez & Martínez-Burgos, 2019*), Geumhong saltpan, South Korea (*Kim et al., 2022*), saltwork of Adamas, Greece (*Lortou et al., 2023*) and Algarve coast, Portugal (*Bombo et al., 2023*). However, to date, there have been no studies seeking this microalga from the island of Mauritius, particularly the southwest region, which is home to several active and abandoned saltpans. Considering the enormous biotechnological potential of this organism, the isolation and identification of cultivable *Dunaliella* strain from unexplored habitats are notable both for research and industrial applications. Furthermore, given that Mauritius aims to increase the share of renewable energy in the country's energy mix by 60% by 2030 (*Republic of Mauritius, 2021*), the utilization of *Dunaliella* as a biofuel feedstock could therefore contribute to achieving this national target.

Set against this background, the present study focuses on isolating and characterizing a salt-tolerant and lipid-producing strain of *Dunaliella*. Additionally, the effect of salinity stress on the growth, biochemical composition, and fatty acid profile of the isolated strain was studied to uncover its potential for biodiesel production.

# MATERIALS AND METHODS

## Water sampling and analysis of physio-chemical parameters

Water samples were collected from a solar saltern (20°22′22.6″S and 57°22′00.4″E) located in the coastal region of Rivière Noire, Mauritius (Fig. S1). Concomitantly with water sampling, environmental parameters such as pH, temperature, electrical conductivity, and dissolved oxygen were recorded *in situ* using a precalibrated multiprobe water quality meter
(YSI, Yellow Springs, OH, USA). Salinity as total dissolved salts (g L$^{-1}$) was estimated gravimetrically by drying 50 mL of water sample at 120 °C for 24 h. The cation (Na$^+$, K$^+$, Ca$^+$ and Mg$^{2+}$) and anion (Cl$^-$ and SO$_4^{2-}$) contents were analyzed by ion chromatography (Dionex$^{TM}$ Aquion$^{TM}$; Thermo Scientific, Waltham, MA USA).

## Single colony isolation and culture maintenance

The water samples were filtered through membranes of decreasing mesh size (200 and 50 μm) to remove unwanted debris (salt, rocks and sand particles) and grazing organisms. The filtrate was then dispensed into 24-well plates and mixed in a 50/50 (v/v) ratio with sterilized modified Johnson medium (*Borowitzka, 1988*) (nutrient composition is shown in Table S1) for the enrichment of microalgal communities. The modified Johnson medium was selected because it is the most commonly used growth medium for the culture of *Dunaliella* (*Borowitzka, 1988*). The inoculated wells were incubated for a 15-day period at 25 ± 1 °C with a light intensity of 50 μmol photon m$^{-2}$ s$^{-1}$ and a photoperiod of 14:10 h light:dark cycle. The wells showing growth were examined microscopically to determine the presence of *Dunaliella* cells.

Following microscopic confirmation, the enriched sample was subjected to a combination of serial dilution and agar plating techniques according to the procedure described by *Olmos-Soto et al. (2012)* with minor modification. Briefly, 1 mL of the enriched sample was diluted serially and 100 μL of the diluted sample from higher dilution (10$^{-5}$) was streaked onto modified Johnson medium supplemented with 1.5% (w/v) bacteriological agar. The inoculated plates were incubated for 15 days under identical culture conditions until visible colonies were observed.

Microalgal samples collected from the natural environment are usually associated with contaminants such as bacteria. To purify the sample, a single algal colony was picked using a sterile inoculation needle and streaked on modified Johnson agar supplemented with a cocktail of antibiotics (100 μg mL$^{-1}$ ampicillin, 50 μg mL$^{-1}$ streptomycin and 15 μg mL$^{-1}$ nalidixic acid). The rationale for using a combination of different antimicrobials is that the contaminating bacteria could have a different sensitivity for each antibiotic (*Droop, 1967*). Modified Johnson agar plates without antibiotics were also inoculated with the same algal colony and used as a control. Both cultures were maintained under the same culture conditions as described above. The axenicity of the cultures was verified by repeated streaking on Marine Agar 2216. Eventually, the axenic culture was kept in sterile antibiotic-free modified Johnson liquid medium and subculture was performed fortnightly to maintain actively growing cells, which were then used as inoculum for subsequent experiments.

## Morphological observations

The morphological observation of the isolate was carried out under a light microscope (DM 1000; Leica Microsystems, Wetzlar, Germany) using 40×–100× magnifications and the image was captured using a digital color camera (DFC 490; Leica Microsystems, Wetzlar, Germany). The isolate was identified based on the morphological description given by *Borowitzka & Siva (2007)*.

## DNA extraction and PCR amplification

Microalgal culture amounting to 10 mL was centrifuged and genomic DNA was extracted from the cell pellet with a Quick-DNA Plant/Seed Miniprep Kit (Zymo Research, Orange, CA, USA) according to the manufacturer's instructions. The 18S rDNA gene and the Internal Transcribed Spacer region were used for DNA fingerprinting and phylogenetic analysis, respectively. The primers used in PCR amplification are listed in Table S2. For both molecular markers, PCR was carried out in a total reaction volume of 50 µL containing 25 µL of OneTaq® PCR master mix (New England Biolabs, Ipswich, MA, UK), 1.0 µL of genomic DNA, 0.2 µL of each primer (100 µM) and 23.6 µL of ultrapure water. The PCR reactions were carried out in a thermocycler (Applied Biosystems, Foster City, CA, USA) using the PCR conditions described by Sathasivam et al. (2012).

The PCR products were visualized on a 1.5% (w/v) agarose gel. A DirectLoad™ 1 kb DNA ladder (Sigma-Aldrich, Burlington, MA, USA) was used as a molecular weight standard.

## Sequencing and phylogenetic analysis

Following amplification, the PCR product obtained with the pair of ITS primers AB1 and AB2 was purified according to the manufacturer's instructions using the GFX™ PCR DNA and Gel Band Purification kit (Cytiva, Marlborough, MA, USA) and sent to Inqaba Biotechnical Industries (Pty) Ltd (Pretoria, South Africa) for sequencing. The resulting ITS-based sequence was inspected with BioEdit (version 7.2.5) software and searched against previously deposited sequences in the GenBank database of the National Center for Biotechnology Information (NCBI) using the Basic Local Alignment Search Tool. Sequences of closely related species were downloaded from GenBank and aligned with those obtained in this work in Geneious Prime software (version 2023.0.1; Biomatters Ltd., Auckland, New Zealand) using the Clustal Omega 1.2.2 plug-in. Phylogenetic relationships were determined using maximum likelihood (ML) and Bayesian inference (BI) methods. The selection of phylogenetic models for the ML method was performed with Molecular Evolutionary Genetics Analysis X software (version 10.1.7) (Kumar et al., 2018). The ML tree was constructed using the Kimura 2-parameter model with discrete gamma distribution (K2+G) and the robustness of the tree topology was determined by bootstrap analysis with 1,000 replicates. BI was performed in the Geneious Prime platform using the Mr. Bayes 3.2.6 plugin. The best substitution models were obtained across the entire general time reversible (GTR) model space. Four simultaneous Markov chains were run for 2,000,000 generations and with a sampling every 1,000 generations. The first 200,000 generations were discarded as burn-in. A majority-rule consensus tree was drawn from the last 1,000 trees. *Chlamydomonas reinhardtii* was chosen as the outgroup.

## Experimental design

The effect of salinity on the microalgal isolate was carried out in batch culture with triplicate systems of 2 L Erlenmeyer flasks containing 1 L of modified Johnson medium. Different

concentrations of NaCl (0.5, 1.0, 2.0, 3.0 and 4.0 M) were added to the culture medium and the initial cell density of the culture was set at approximately $10 \times 10^5$ cells mL$^{-1}$. The light condition and temperature were maintained at 50 µmol photons m$^{-2}$ s$^{-1}$ with a photoperiod of 14:10 h of light:dark cycle and $25 \pm 1\,°C$, respectively. All flasks were shaken three times daily.

## Determination of the optimal salinity for growth

The salinity tolerance of the isolate was determined by studying its cell growth at five different concentrations of NaCl for a period of 35 days. The cell count of each flask was performed every two days using a Neubaur-improved counting chamber and light microscope (DM1000; Leica Microsystems, Wetzlar, Germany). In short, approximately 1 mL of culture was taken and fixed with 100 µL of Lugol's iodine solution. Then 100 µL of fixed sample was introduced into the cell counting chamber slide and the cell count was performed at 40× magnification. The growth curves were plotted at the end of the experiment using the cell count data. The specific growth rate (µ) and the doubling time (k) were calculated according to the following equations adapted from *Wood, Everroad & Wingard (2005)*:

$$\mu = \frac{\ln\left(\frac{N_2}{N_1}\right)}{t_2 - t_1} \qquad (1)$$

$$k = \frac{\ln 2}{\mu} \qquad (2)$$

where $N_1$ and $N_2$ are the number of cells at time $t_1$ and $t_2$, respectively.

## Quantification of carbohydrates, proteins, lipids, and photosynthetic pigments

To determine the biochemical composition, microalgal cells exposed to different salinities were harvested in the stationary phase. For this procedure, the microalgal culture was centrifuged at 6,000 rpm for 10 min and the supernatant was discarded. The pellet was washed twice with 100 mL of 0.5 M ammonium formate to remove the residual salt and subsequently dried in an oven at 60 °C for 7 h according to *Hosseinizand, Sokhansanj & Lim (2018)*. The dried biomass was finely ground using an agate pestle and mortar and stored at −20 °C prior to the extraction of carbohydrate, protein, lipids and pigments.

The total carbohydrate content was determined using the phenol–sulfuric acid method, as previously described by *Singh et al. (2019a)*. Briefly, 3 mL of 72% (w/w) sulfuric acid was added to 50 mg dried biomass and incubated at 30 °C for 0.5 h. The hydrolysate was then diluted to 4% (w/w) sulfuric acid with distilled water and autoclaved at 121 °C for 20 min. After autoclaving, the hydrolyzed sample was neutralized with sodium carbonate. The content was then centrifuged and the supernatant (0.1 mL) was used to determine the total carbohydrate content by following the method of phenol sulfuric by *DuBois et al. (1956)*.

The protein content was calculated using the nitrogen-protein conversion factor of 4.78 (*Lourenço et al., 2004*). The elemental nitrogen content of the microalgal biomass was measured using an elemental analyzer (Euro EA 3000; Germany). Lipid extraction was carried out using the method described by *Kim, Lee & Lee (2016)*. In short, 50 mg of dried biomass was suspended in a mixture of chloroform, methanol and water (ratio 1:2:0.8, v/v) and subjected to ultrasonic wave (for 2 min–30 s On and 15 s Off, at 20 kHz) to achieve cell disruption. The content was centrifuged at 5,000 rpm for 5 min and the bottom organic phase containing the lipids was transferred to preweighted glass vial. The aqueous phase containing the microalgal biomass was extracted one more time with same mixture of chloroform, methanol and water. The pooled organic phases were evaporated at 65 °C in an oven until a constant weight of lipid was achieved. The total lipid content was determined according to Eq. (3):

$$Lipid\ content\ (\%) = \frac{W_2 - W_1}{M} \times 100 \tag{3}$$

where $W_1$ is the weight of the empty vial (g), $W_2$ is the weight of the vial with the dried lipid extract (g) and M is the weight of the dried algal biomass (g).

Chlorophyll *a* and total carotenoids content were quantified according to the methodology proposed by *Santhakumaran, Kookal & Ray (2018)*. Briefly, 20 mg of algal sample was homogenized in darkness using 20 mL of chilled 80% acetone. The homogenate obtained was centrifuged at 6,000 rpm for 10 min at 15 °C. After centrifugation, the supernatant was collected, and the residue was again homogenized with 80% acetone and this process was carried out until the sample became colorless. The pooled supernatants were made up to 100 mL in a volumetric flask with 80% acetone and the absorbance was measured at 480, 510, 645 and 663 nm. The pigment content was calculated using Eqs. (4) and (5) and expressed as mg g$^{-1}$:

$$Chlorophyll\ a = \frac{12.7\ \times\ A_{663} - 2.69\ \times A_{645}\ \times volume\ of\ sample\ (mL)}{1000\ \times weight\ of\ algal\ biomass\ (g)} \tag{4}$$

$$Total\ carotenoids = \frac{7.6\ \times A_{480} - 1.49\ \times A_{510} \times volume\ of\ sample\ (mL)}{1000\ \times weight\ of\ algal\ biomass\ (g)} \tag{5}$$

where $A_{663}$, $A_{645}$, $A_{510}$ and $A_{480}$ is the absorbance measured at 663, 645, 510 and 480 nm, respectively, using a UV-visible spectrophotometer (Jenway 7305, UK).

## Fatty acid profiling through analysis of fatty acid methyl ester

To study the fatty acid profile, a direct transesterification step of the microalgal lipid was performed using the protocol of *Moll et al. (2014)*. The resulting fatty acid methyl ester (FAME) samples were analyzed using a Shimadzu GC-2014 gas chromatograph equipped with a flame ionization detector. A fused silica capillary column SP-2560 (Supelco, 100 m × 0.25 mm internal diameter, 0.25 mm film thickness) was used along with helium as carrier gas. The column oven temperature was initially held at 140 °C for 5 min, followed by an

increase to 240 °C at rate of 4 °C min$^{-1}$ and kept at that temperature for 5 min. The injector and detector temperatures were set at 260 °C. The FAMEs were quantified by comparing their retention times with an external fatty acid standard (Supelco 37 Component FAME Mix; Sigma-Aldrich, Burlington, MA, US). The fatty acid composition was expressed as a relative mass percentage (mass %) of the total fatty acid mixture.

### Determination of biodiesel quality parameters

Important biodiesel quality parameters such as the average degree of unsaturation (ADU), kinematic viscosity (KV), density (ρ), cloud point (CP), cetane number (CN), iodine value (IV), higher heating value (HHV), long-chain saturated factor (LCSF), filter plugging point (CFPP), oxidative stability (OS) and flash point (FP) were calculated based on the fatty acid composition of algal lipids using empirical Eqs. (6)–(16) built by *Ramos et al. (2009)* and *Hoekman et al. (2012)*.

$$ADU = \sum N \times Mf \tag{6}$$

$$KV = -0.6313\ ADU + 5.2065 \tag{7}$$

$$p = 0.0055\ ADU + 0.8726 \tag{8}$$

$$CP = -13.356\ ADU + 19.994 \tag{9}$$

$$CN = -6.6684\ ADU + 62.876 \tag{10}$$

$$IV = 74.373\ ADU + 12.71 \tag{11}$$

$$HHV = 1.7601\ ADU + 38.534 \tag{12}$$

$$LCSF = (0.1 \times C16:0) + (0.5 \times C18:0) + (1 \times C20:0) + (1.5 \times C22:0) \\ + (2 \times C24:0) \tag{13}$$

$$CFPP = (3.1417 \times LCSF) - 16.477 \tag{14}$$

$$OS = 117.9295X + 2.5905 \tag{15}$$

$$FP = 205.226 + (0.083 \times C16:0) - (1.723 \times C18:0) - (0.5717 \times C18:1) \\ - (0.3557 \times C18:2) - (0.46 \times C18:3) - (0.2287 \times C:22) \tag{16}$$

where N is the number of double carbon-carbon bonds in a fatty acid, and Mf is the main fraction of each fatty acid. C16:0, C18:0, C20:0, C22:0 and C24:0 are the weight percentages of each of the fatty acids expressed in wt.%. X is the content of linoleic (C18:2) and linolenic acids (C18:3) (wt.%).

## Statistical analysis

In this study, data (except for the FAME profile) are expressed as mean ± standard deviation (SD). Differences between mean values were calculated using Tukey's test at 0.05 significance level. Statistical analyzes and graphs were created using GraphPad Prism 9.0 software (GraphPad, San Diego, CA, USA).

# RESULTS AND DISCUSSION

## Isolation and morphological characterization

Establishing pure microalgae cultures through an isolation process represents the first step in the development of any microalgae-based technologies. Despite the availability of live microalgae cultures from culture collections worldwide, the selection of native strains is preferable due to their superior adaptability to local climatic conditions and low biosecurity risks. Solar salterns are often subjected to extreme abiotic conditions, including osmotic stress due to salt build-up, low dissolved oxygen, and high solar irradiance. Therefore, such an extreme environment is anticipated to be an ideal source for obtaining robust strains of microalgae that have naturally adapted to grow in extreme salinity. Furthermore, microalgae originating from an environment that undergo harsh or fluctuating conditions are assumed to accumulate a greater amount of energetic storage compounds such as lipids, which is the main raw material used in biodiesel production (*Duong et al., 2012*). Therefore, in this study, water sampling was carried out from a solar saltern located in the coastal region of Rivière Noire, Mauritius. The physiochemical characteristics of the saltpan water samples are detailed in Table S3. The pond from which the samples were collected had a temperature of $33.7 \pm 0.1\,^{\circ}C$ and a pH of $7.6 \pm 0.03$. The salinity of the water at the time of sampling was $262.1 \pm 7.9$ g $L^{-1}$ (4.4 M), indicating a hypersaline medium. The mean dissolved oxygen and electrical conductivity were $2.25 \pm 0.03$ mg $L^{-1}$ and $152.52 \pm 13.67$ ms $cm^{-1}$, respectively. The chemical composition of the water samples was dominated by sodium and chloride ions followed by sulfate and magnesium ions.

Several isolation techniques including fluorescence-activated cell sorting, micromanipulation, serial dilution, and streak plating have been developed to recover microalgae from their natural environment. In this study, the serial dilution and agar plating method were used for the isolation process. This methodology has been successfully applied in previous studies involving green microalgae such as *Chlorella* and *Nannochloropsis* (*Oyebamiji et al., 2019*; *Rani et al., 2022*). Following the process of serial dilution and streak plating, a single microalgal isolate (designated as SCH-18) was successfully isolated. The isolate grew and survived on modified Johnson medium agar by forming green homogenous colonies. The algal colony was axenized using a combination of antibiotics, namely ampicillin, streptomycin, and nalixidic acid. The selection of these antimicrobials was based on previous work in which the successful elimination of bacterial contaminants eukaryotic microalgae cultures was reported (*Wang et al., 2016*; *Olmos-Soto et al., 2012*). Many authors have highlighted the importance of obtaining axenic algal culture (*Shiraishi, 2015*; *Vu et al., 2018*). In fact, culture without contaminants

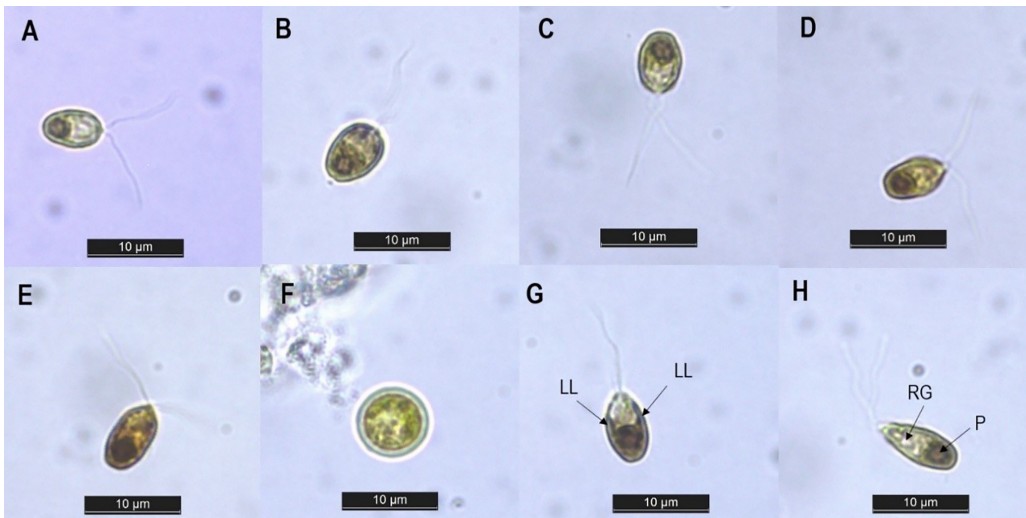

**Figure 1 Light microscopy image of *Dunaliella* isolate SCH-18 under 0.5 M (A), 1.0 M (B), 2.0 M (C), 3.0 M (D) and 4.0 M (E) salt conditions, taken at 100x magnification. Image (F) corresponds to aplanospore. Images (G and H) depict the presence of lateral lobes (LL), refractile granules (RG) and pyrenoid (P).** Bar = 10 μm.

is essential to ensure that experimental results, particularly in studies related to gene sequencing, algal physiology, and biochemistry, are not affected by irrelevant microorganisms (*Shiraishi, 2015*).

The isolated strain was then identified by studying its morphological characteristics under a light microscope. The cell morphology of our isolate under different concentrations of NaCl is presented in Figs. 1A–1E. Along the different salinities, the cells were unicellular, radially symmetrical and their color was always green, but their shape varied from ellipsoid at NaCl levels of 0.5–2.0 M to ovoid at salinities of 3.0–4.0 M. Similar alteration in cell shape along various salinity regimes has been previously reported in *Dunaliella* cultures by *Park et al. (2015)* and *Singh et al. (2019b)*. This morphological plasticity is due to the absence of a rigid cell wall in *Dunaliella* species. This lack of cell wall allows the cells to change their shape and volume according to the osmotic state of their surrounding environment (*Shetty, Gitau & Maróti, 2019*). The cell dimensions recorded along the different salinities ranged from 5.56–9.71 μm in length and 3.79–5.98 μm in width ($n = 30$) (Table 1). Exposure to rising salinity led to an increase in cell length, width and volume. Cell volume increased from 65.39 μm$^3$ at 0.5 M NaCl to 114.12 μm$^3$ at 4.0 M NaCl. This hike in cell size may be attributed to the accumulation of the compatible solutes notably glycerol, which has been reported to exceed 50% of dry weight when grown at salinity of 4.0 M NaCl and above (*Oren, 2017*). Similarly, *Singh et al. (2019b)* recorded an increase in the intracellular glycerol content of *Dunaliella salina* with increasing salinity. With respect to cell structure, two equal flagella were attached to the apical area, with a length in the range of 11.98–13.21 μm. The length of the flagellum was found to be equal to 1.5 to two times the length of the cell. A single cup-shaped chloroplast occupying approximately half the cell volume and containing a large pyrenoid surrounded by distinct amylosphere (as illustrated in Fig. 1H) was observed in the basal region of the cells. The

**Table 1 Dimension of cells recorded along the different salinities.**

**Cell dimensions**

| Salinity (M NaCl) | Cell length range (μm) | Average of cell length (μm) | Cell width range (μm) | Average of cell width (μm) | Flagella length (μm) | Flagella to cell length ratio | Cell volume (μm³) |
|---|---|---|---|---|---|---|---|
| 0.5 | 5.56–7.47 | 6.45 ± 0.50 | 3.79–4.84 | 4.38 ± 0.32 | 12.93 ± 0.58 | 2.00 | 65.39 ± 12.55 |
| 1.0 | 5.60–7.40 | 6.97 ± 0.49 | 4.06–5.98 | 5.03 ± 0.49 | 11.98 ± 0.28 | 1.72 | 93.08 ± 18.53 |
| 2.0 | 6.22–8.15 | 7.04 ± 0.51 | 4.25– 5.19 | 4.59 ± 0.33 | 12.40 ± 0.77 | 1.76 | 78.11 ± 12.18 |
| 3.0 | 6.20–9.57 | 8.01 ± 0.91 | 4.13–4.93 | 4.52 ± 0.26 | 12.07 ± 0.84 | 1.51 | 86.56 ± 16.52 |
| 4.0 | 7.80–9.71 | 8.75 ± 0.71 | 4.06–5.93 | 4.96 ± 0.46 | 13.21 ± 0.76 | 1.51 | 114.12 ± 26.36 |

lateral lobes of the chloroplast did not reach the anterior end of the cell (Fig. 1G). Stigma was not detected while there were refractile granules in the anterior portion of the cells (Fig. 1G). No sexual stages were observed, but spherical aplanospores with a two-layered smooth wall and mean diameter of 7.09 ± 0.97 μm were visible at salinities of 2.0 M and above (as shown in Fig. 1F). Based on the taxonomic descriptors presented by *Borowitzka & Siva (2007)*, our isolate showed several features similar to the genus *Dunaliella* and resembled the description of *Dunaliella tertiolecta* (belonging to section *Tertiolectiae*) in terms of cell shape and size, ratio of cell length to flagella length, and presence of refractile granules. However, species identification based solely on morphology can lead to false conclusions, since the morphological characteristics of *Dunaliella* can vary with culture conditions and the developmental stage of cells (*Polle, Struwe & Jin, 2008*). In view of this, phylogenetic analysis of the internal transcribed spacer (ITS) region was performed to supplement morphological observation. This region of nuclear ribosomal DNA is believed to be an excellent marker for delineating strains at the species level (*Polle, Struwe & Jin, 2008*; *Borovkov et al., 2021*).

## Phylogenetic analysis and 18S rDNA intron sizing

The amplified ITS gene of the isolate was 650 base pairs in size. The ITS sequence was then aligned with previously deposited sequences in the NCBI database. The phylogenetic tree analysis revealed four distinct clades (as shown in Fig. 2). The first clade (clade A) comprised the isolate SCH18, three strains of *Dunaliella tertiolecta* (CCAP 19/6B, Dtsi and VM), two strains of *Dunaliella primolecta* (hd9 and CCAP11/34), two strains identified at the genus level (ABRIINW-S6.3 and ABRIINW-G22) and one strain of *Dunaliella quartolecta* (CCAP 19/8), *Dunaliella bioculata* (UTEX 199), *Dunaliella minuta* (CCAP 19/5), *Dunaliella polymorpha* (CCAP 19/7A), *Dunaliella bardawil*, *Dunaliella maritima* (SAG 42.89), *Dunaliella peircei* (UTEX 2192) and *Dunaliella parva*, respectively. This clade showed moderate statistical support in maximum likelihood and Bayesian analyzes (67 BS/0.83 PP). The second, third, and fourth clades (clade B, C, and D) which were supported by high bootstrap values, consisted exclusively of strains of *Dunaliella salina*, *Dunaliella parva*, and *Dunaliella virdis*, respectively. According to the phylogenetic tree, the nearest neighbor of our isolate SCH18 was *Dunaliella* sp. ABRIINW-G22 (single base pair difference) and *Dunaliella* sp. ABRIINW-S6.3 (two base pair difference), both of which were isolated from a

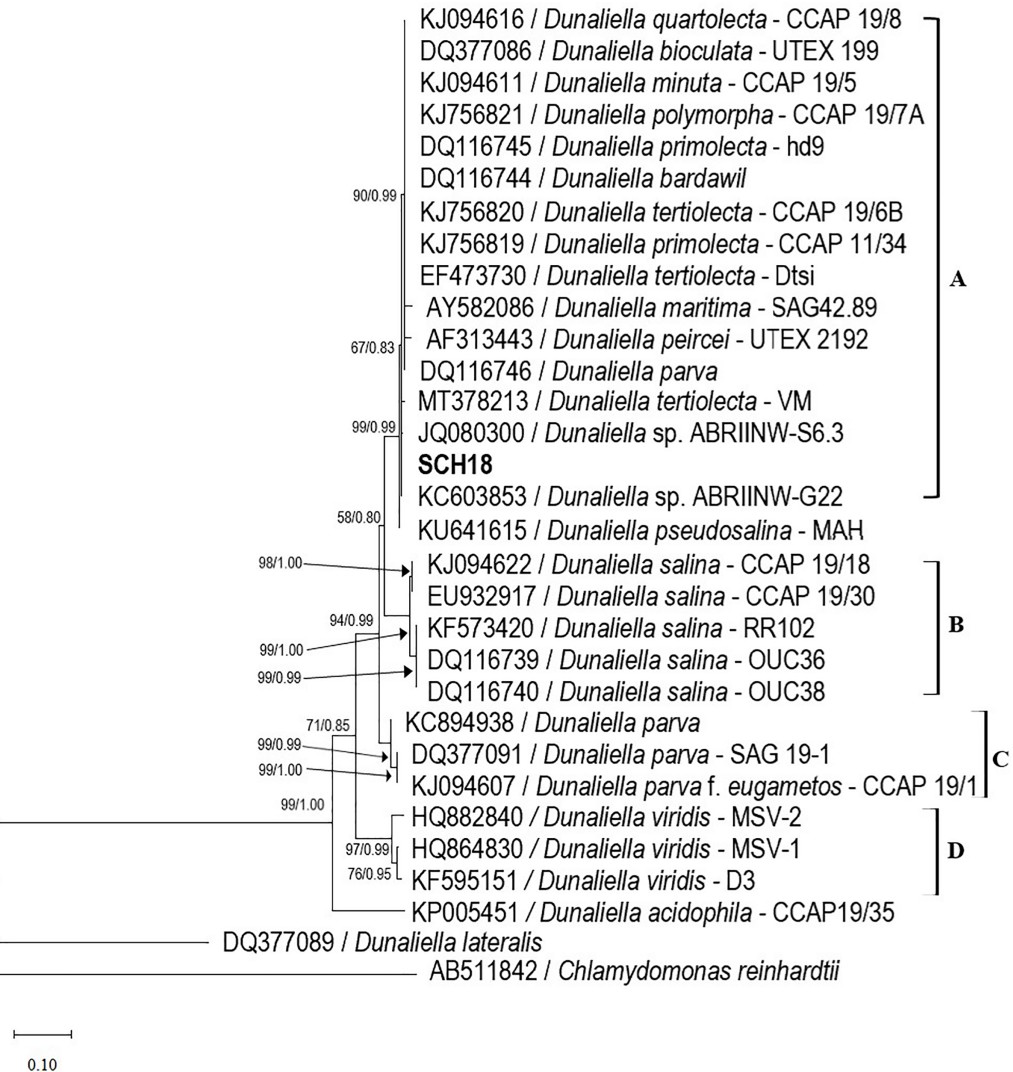

**Figure 2 Phylogenetic tree of *Dunaliella* species inferred from the ITS region based on maximum likelihood and Bayesian inference analyzes.** The values at the nodes represent bootstrap support and Bayesian posterior probability support (ML/BI). The bold type indicates the local isolate used in this study. The scale bar represents the number of nucleotide substitutions per site. The accession number of strain SCH18 is OR158048.

salt marsh in Iran. The nucleotide differences between our isolate and the remaining species of Clade A ranged from three to six base pairs, indicating that most of these strains had almost similar ITS sequences. Other researchers noted similar results when performing a phylogenetic analysis of the genus *Dunaliella* (*Assunção et al., 2012*; *Emami et al., 2015*). The appearance of several strains with identical ITS sequences may be a consequence of misidentification. In fact, various authors have mentioned that several *Dunaliella* strains available in culture collections carry misleading names, since their taxonomic assignments were previously based on morphological traits (*Borowitzka & Siva, 2007*). To this end, the taxonomic status of many reported *Dunaliella* strains was revised by *Assunção et al. (2012)*, based on their ITS2 sequences and secondary structures. The authors proposed

reclassification of the following sequences-*D. quartolecta* CCAP19/8, *D. bioculata* UTEX199, *D. minuta* CCAP 19/5, *D. primolecta* hd9 (accession number DQ116745), *D. bardawil* (accession number DQ116744), *D. maritima* SAG 42.89, *D. peircei* UTEX 2192 and *D. parva* (accession number DQ116746), as *D. tertiolecta*. On the basis of this reclassification, our isolate was found to be phylogenetically closely related to strains of *D. tertiolecta*, which is consistent with the morphological data. The DNA sequence data is publicly available in GenBank with accession number OR158048.

The identity was further validated with the intron-sizing method, which focuses on the size of the 18S rDNA gene, and the sequence variability presented by the group 1 introns of *Dunaliella* species (*Olmos Soto, 2015*). In this study, the size and occurrence of introns within the 18S rDNA gene of the isolate SCH18 were determined using a set of conserved primers (MA1, MA2, and MA3) and species-specific primers (DSs, DBs, and DPs), respectively. PCR amplification with the MA1-MA2 and MA1-MA3 primer pairs produced two identical DNA bands of approximately 1,700 bp (as depicted in Fig. S2), which is close to the reported size for *D. tertiolecta* (*Olmos-Soto et al., 2012*). Amplification with species-specific primers (which are directed to the introns of 18S rDNA) did not give any amplicon on the electrophoresis gel, which confirms the absence of a group I intron in our isolate. According to *Olmos-Soto et al. (2012)*, the lack of an intron in the 18S rDNA gene shows the characteristics of *D. tertiolecta*, while the presence of an intron (750 bp) characterizes *D. salina* and two introns (1,000 bp) characterize *D. parva* and *D. bardawil*.

## Optimum salinity for growth

The optimal salinity for the growth of the *Dunaliella* isolate SCH18 was evaluated by growing the cells at five different concentrations of NaCl. Its growth performance was monitored by measuring cell density. The result represented in Fig. 3A showed that the isolate was able to grow in all the salt concentrations tested and, for each salt regime, a lag phase was observed during the first three days. This lag phase can be attributed to the acclimatization of the microalga to its new saline environment. The acclimatization period was followed by an exponential phase which was reflected in the growth curves as an increase in cell density from day 4 to day 30. On day 30 cell density values of $5.72 \pm 0.09 \times 10^7$ and $4.64 \pm 0.08 \times 10^7$ cell mL$^{-1}$ were achieved in cultures with salt concentrations of 1.0 M and 0.5 M, respectively. At salinities of 2.0 and 3.0 M, moderate cell densities were recorded, with an average of $4.42 \pm 0.01 \times 10^7$ and $3.86 \pm 0.10 \times 10^7$ cell mL$^{-1}$ on day 30, respectively. The cells grew and survived in salinity of 4.0 M, but a relatively lower cell density ($1.62 \pm 0.01 \times 10^7$ cell mL$^{-1}$) was obtained.

As can be seen from Fig. 3B, a maximum growth rate of $0.39 \pm 0.003$ day$^{-1}$ and a short doubling time of $1.79 \pm 0.01$ days were achieved with salinity of 1.0 M. It could also be observed that exposure to increasing salinities led to a reduction in the specific growth rate and an increase in the doubling time. For example, at salinity of 4.0 M, the growth rate was at a minimum ($0.11 \pm 0.006$ day$^{-1}$) and the cells had the longest doubling time ($6.24 \pm 0.35$ days). This reduction in growth rate can be explained by the fact that, at higher salinities, microalgae tend to spend most of their energy on osmoregulatory mechanisms to protect against oxidative injuries and to equilibrate their internal osmotic pressure with the

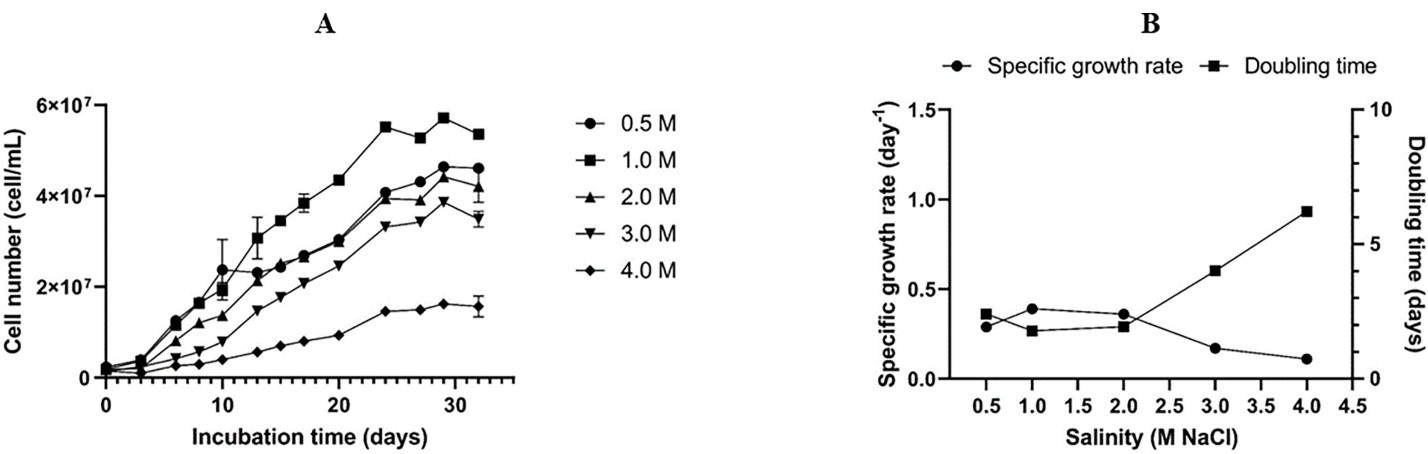

**Figure 3** Growth curves (A), specific growth rate, and doubling time (B) of the *Dunaliella* isolate SCH18 grown at different salinities.

surrounding salinity rather than increase their growth. Typically, *Dunaliella* uses a multitude of osmoregulatory mechanisms to cope with salt stress, including drastic changes in morphology, modification of their photosynthetic pigment composition, up-regulation of heat shock proteins, and genes related to starch hydrolysis and glycerol synthesis (*Keil et al., 2023*).

Results from the growth studies showed that salinity of 1.0 M was optimal for achieving a high growth rate. Normally, a high growth rate translates into greater biomass productivity. On the contrary, suppressed cell growth was observed at salt concentration of 4.0 M. Our results also confirmed that the *Dunaliella* isolate SCH18 can tolerate and grow in a certain range of salt concentrations (0.5 to 3.0 M NaCl). This physiological characteristic is typical of the section *Tertiolectiae*, which contains species that are euryhaline in nature. Species in the *Tertiolectiae* section are known to grow best in salt concentrations less than 1.0 M (<6% NaCl) but are tolerant to salt saturation of up to 5.5 M (34% NaCl) (*González, Gómez & Polle, 2019*). For instance, the oligo-euhaline strain of *D. tertiolecta* has been reported to grow at NaCl concentrations ranging from 0.17 to 3.5 M (*Park et al., 2015*). Euryhaline microalgae, such as the one isolated in this study, could be a promising candidate for large scale cultivation in open-pond systems because of its tolerance to a wide range of salinities. Furthermore, for an island state like Mauritius, the cultivation of microalgae that can accommodate saline conditions is more attractive due to easy and direct access to seawater.

## Effect of different salinities on photosynthetic pigment composition

The amount of chlorophyll *a* and total carotenoids in the *Dunaliella* isolate SCH18 cultured under different concentrations of NaCl was determined to obtain information on its carotenogenic activity. The ability to hyperaccumulate carotenoids has previously been used as a classification tool to differentiate *Dunaliella* spp. In this study, the chlorophyll *a* content varied from 5.89 to 9.45 mg g$^{-1}$ (as shown in Table 2). The highest chlorophyll *a*

**Table 2 Effects of NaCl concentrations on content of chlorophyll *a* (Chl *a*), total carotenoids content (Car), and total carotenoids to chlorophyll *a* ratio (Car/Chl *a*).**

| NaCl concentration (M) | Chl *a* (mg g$^{-1}$) | Car (mg g$^{-1}$) | Car/Chl *a* ratio |
|---|---|---|---|
| 0.5 | 7.36 ± 0.46 | 2.75 ± 0.64 | 0.37 |
| 1.0 | 9.45 ± 0.21 | 3.21 ± 0.06 | 0.34 |
| 2.0 | 8.71 ± 0.42 | 2.11 ± 0.15 | 0.24 |
| 3.0 | 8.51 ± 1.61 | 1.82 ± 0.07 | 0.21 |
| 4.0 | 5.88 ± 0.95 | 1.54 ± 0.19 | 0.26 |

content was recorded in cells grown under 1.0 M salinity, while the lowest amount of this pigment was observed in 4.0 M. This result is in agreement with the cell density value of this study, since chlorophyll *a* content is known to be directly related to the concentration of algal cells. Similarly to our findings, *Park et al. (2015)* observed a decrease in the chlorophyll content of *D. tertiolecta* due to salt stress. The authors also found that *D. salina*, a well-recognized carotenogenic species, was able to maintain its total chlorophyll content over a wide range of salinities (0.8–4.5 M). In another study carried out by *Tammam, Fakhry & El-Sheekh (2011)*, a similar trend in chlorophyll content was reported for *D. tertiolecta* with the highest value recorded at 1.0 M. On the contrary, *D. salina* was found to maintain a higher chlorophyll concentration in a wide salinity ranging from 0.05–4.0 M. This reduction in the content of photosynthetic pigment due to hypersaline condition may be regarded as a protective strategy to reduce the probability of photooxidative damage (*Park et al., 2015*). Another possible explanation for this reduction could be due to the high concentration of Na$^+$ ions, which is believed to incite the overproduction of reactive oxygen species, which in turn is known to induce lipid peroxidation of the thylakoid membranes and cause progressive damage to the photosystem II reaction center (*Zhang et al., 2018*).

Carotenoids are lipophilic compounds that act as an accessory pigment in microalgae. In addition to their light-harvesting functions in the photosystems, carotenoids are also involved in the protection of microalgal cells against photooxidative damage by serving as scavengers of stress-induced reactive oxygen species. The content of total carotenoids recorded in this study ranged from 1.54 to 3.21 mg g$^{-1}$ (as shown in Table 2). The maximum level of total carotenoids was found in 1.0 M NaCl, while the minimum level of this pigment was found in 4.0 M NaCl. This is in agreement with *Tammam, Fakhry & El-Sheekh (2011)*, who reported a high level of β-carotene content in *D. tertiolecta* when exposed to a salinity of 1.25 M. Similarly to our findings, the latter also observed a decrease in the cellular β-carotene content with increasing salinity. In *Dunaliella*, hyperaccumulation of carotenoids, mainly in the form of β-carotene, has been reported to occur under conditions that are unfavorable for growth, such as high salinity, strong light intensity, and low nitrogen concentration (*Borowitzka & Siva, 2007*). However, this ability to hyperaccumulate carotenoids is restricted to only four species, namely *D. parva*, *D. salina*, *D. pseudosalina* and *D. bardawil*. For example, the total carotenoids concentration

in *D. salina* AthU-AI D31 was found to reach 26.65 mg g⁻¹ dry cell weight when cultured under salt stress of 2.73 M NaCl (*Chantzistrountsiou et al., 2023*).

Carotenoids to chlorophyll *a* ratio has been considered a presumptive indicator of carotenogenesis in microalgae (*Solovchenko et al., 2013*). Likewise, ratio of carotenoids to chlorophyll *a* of a carotenogenic strain of *D. salina* cultivated at different salinities (0.25–4.0 M) was found to increase from 046 to 0.76 with increasing NaCl concentration (*Singh et al., 2019b*).Similar observation was made by *Hadi, Shariati & Afsharzadeh (2008)* for *D. salina* grown under different salinities. However, in our study, the opposite trend was observed with the ratio of carotenoids to chlorophyll *a*, decreasing from 0.37 to 0.26 with increase in NaCl concentrations (as shown in Table 2). This result suggests that our isolate is not able to overproduce carotenoids like many *D. tertiolecta* strain previously reported. The mechanism behind β-carotene hyperaccumulation in *Dunaliella* has recently been deciphered by *Kim et al. (2024)* through a comparison of *D. salina* and *D. tertiolecta*.

Thus, it is confirmed that the isolated strain SCH18 belongs to the species *D. tertiolecta* based on its morphological, physiological and molecular characteristics.

## Effect of different salinities on biochemical composition

In Table 3 the biochemical composition of *D. tertiolecta* SCH18 grown under different salinities is shown. The protein content was found to increase as the salt concentration in the culture medium was increased from 0.5 to 1.0 M, and beyond this, the protein content decreased with increasing salinity. The highest amount of protein (30.35%) was obtained in biomass grown at 1.0 M salinity and the lowest (18.48%) under 4.0 M salinity (as shown in Table 3). Similarly to our findings, green microalgae, such as *Chlamydomonas reinhardtii* and *Acutodesmus dimorphus*, have shown decreased protein under salt stress (*Chokshi et al., 2017*; *Fal et al., 2022*). *Dittami et al. (2011)* related this reduction in protein levels during salt stress to the downregulation of genes associated with protein synthesis and the activation of genes related to protein degradation. Amino acids released from protein breakdown during salt stress have been reported to serve in the elimination of reactive oxygen species and the maintenance of the metabolic status of stressed cells (*Singh et al., 2019b*). The protein content value in this study was within the range of 8.3–31.3% as reported by *Efremenko et al. (2012)* for *D. tertiolecta*. In another study, the protein content of *D. tertiolecta* cultivated in three different media with salinity of 0.6 M showed considerable variation (19.43–42.19%) (*Ahmad, Amin & Ashraf, 2024*). Although the values reported by *Ahmad, Amin & Ashraf (2024)* are higher than those reported in this study, it is worth highlighting that the culture conditions, drying process and the quantification method differ from those used in our study and may therefore influence the different results. Nonetheless, the protein content of our *D. tertiolecta* isolate still shows higher value than some traditional sources of animal protein such as beef (17.4%), fish (19.2–20.6%), and chicken (19–24%) (*Koyande et al., 2019*). Research has shown that the protein content of *Dunaliella* has a well-balanced amino acid profile and that the absence of a rigid cell wall makes it more digestible by both humans and animals (*Sui & Vlaeminck, 2020*). Therefore, it can be used as a protein supplement for humans and animal nutrition.

**Table 3 Biochemical composition (percent dry cell weight, % DCW) of the *Dunaliella* isolate SCH-18 grown under different NaCl concentrations (0.5–4.0 M).**

| NaCl concentration (M) | Total proteins (% DCW) | Total lipids (% DCW) | Total carbohydrates (% DCW) |
| --- | --- | --- | --- |
| 0.5 | 21.45[bc] ± 2.73 | 15.71[ab] ± 0.89 | 53.14[a] ± 2.89 |
| 1.0 | 30.35[a] ± 0.18 | 17.81[a] ± 2.40 | 42.02[b] ± 5.20 |
| 2.0 | 26.26[ab] ± 1.42 | 16.96[a] ± 0.54 | 27.87[c] ± 1.25 |
| 3.0 | 24.50[b] ± 1.78 | 16.17[a] ± 1.13 | 25.68[c] ± 2.04 |
| 4.0 | 18.48[c] ± 2.16 | 12.67[b] ± 0.48 | 18.67[d] ± 2.51 |

Note:
Means (±SD) with different superscripts in the same column are significantly different ($p < 0.05$).

Regarding lipids, intracellular content increased from 15.71 to 17.81% when salinity was raised from 0.5 to 1.0 M but decreased to 12.67% when 4.0 M NaCl was applied (as shown in Table 3). There were no significant differences in total lipid in cells grown at a salinity level of 1.0 to 3.0 M, suggesting that salinity variances, which are normally expected in open outdoor ponds, will have an insignificant effect on the lipid content of the organism. These data are in agreement with those of *Ahmed et al. (2017)* and *Araj-Shirvani et al. (2024)*, who reported that the lipid content within the *Dunaliella* genus varies from 6% to 71%. Likewise, *Chen et al. (2012)* and *Ahmad, Amin & Ashraf (2024)*, respectively, reported a low lipid content of 17.80% and 15.32–18.14 % in *D. tertiolecta*. In another study by *Khatoon et al. (2017)*, the highest lipid content of 14.9% was obtained in *Dunaliella* sp. when grown in salinity of 10 ppt compared to 30 ppt and 50 ppt. Although the lipid yield of our *D. tertiolecta* isolate SCH18 was lower compared to some traditional biodiesel feedstocks such as soybean (19.2%), canola seed (40.5%), sunflower seed (41.9%) (*Fawcett et al., 2022*), the lipid content was found to be higher than that reported in the literature for other microalgae (*Almutairi, 2022*; *Mathimani et al., 2023*). To ensure the commercial success of microalgae-based biodiesel production, it is important to achieve a higher lipid yield. Reports have shown that the lipid content of microalgae can be improved by optimizing cultivation conditions and manipulating the availability of nutrients (*Babu et al., 2022*).

In the case of carbohydrate content, a maximum production of 53.14% and 42.02% were obtained with 0.5 M and 1.0 M NaCl. Above this salinity, carbohydrate production was greatly attenuated, reaching 18.67% at 4.0 M NaCl (as shown in Table 3). In another study with *D. tertiolecta*, a total carbohydrate content of 46.5–56.8% was obtained by *Efremenko et al. (2012)*. Therefore, the result of the present study provides evidence that the isolated microalga has a substantial carbohydrate content, which can be used as feedstock for the bioethanol industry. Previous studies have shown that *D. tertiolecta* carbohydrate is mainly composed of glucose, which is considered one of the primary raw materials in the production of bioethanol (*Markou, Angelidaki & Georgakakis, 2012*). Therefore, by means of a biorefinery concept, the algal biomass thus obtained can be used to produce biodiesel and bioethanol as a coproduct.

Several studies have investigated and reported that abiotic stress conditions, such as high salinity, stimulate a higher accumulation of carbohydrates and lipids in microalgae,

compared to proteins (*Chokshi et al., 2017*; *Ji et al., 2018*; *Fal et al., 2022*). However, in this study, both the carbohydrate and lipid content were found to decrease with increasing salinity. The reason for this observation could be due to the reallocation of carbon for the biosynthesis of glycerol, which acts as an osmolyte and helps to maintain cell turgor. When exposed to extreme salinity, glycerol production in *Dunaliella* can reach 50% of total dry weight (*Oren, 2017*). According to *Goyal (2007)*, the carbon molecule needed for glycerol biosynthesis is derived mainly from starch degradation. Since starch is also the main carbon source for lipid biosynthesis in *Dunaliella* (*Pick & Avidan, 2017*), a decrease in lipid yield is thus expected with a reduction in starch level during salt stress.

## Effect of different salinity on the fatty acid profile

The fatty acid composition of the isolate SCH18 cultured at different concentrations of NaCl is presented in Table 4. In all treatments, the total fatty acid content was mainly composed of myristic acid (C14:0), palmitic acid (C16:0), palmitoleic acid (C16:1) stearic acid (C18:0), oleic acid (C18:1), linoleic acid (C18:2), gamma ($\gamma$) linolenic acid (C18:3n-6) and alpha ($\alpha$) linolenic acid (C18:3n-3). Previous studies have reported that lipids containing fatty acids with carbon chain length numbers between 16 and 18, are the most desirable feedstock for biodiesel synthesis (*Suastes-Rivas et al., 2020*).

In this study, the content of polyunsaturated fatty acids (PUFA) accounted for the highest proportion (34.5% to 47.4%) followed by saturated fatty acids (SFA) (18% to 25.1%) and monounsaturated fatty acids (MUFA) (5.1% to 6.7%). The increase in salinity from 0.5 to 4.0 M NaCl led to an increase in the contents of SFA and MUFA. This increment was driven primarily by myristic acid, palmitic acid, and oleic acid, respectively. On the contrary, the percentage of PUFAs, particularly linolenic acids, was found to decrease with an increase in salt concentrations. This alteration in the fatty acid pattern during salt stress is consistent with previous studies where a similar increase SFA and MUFA levels and a decrease in the content of PUFA were reported in *Dunaliella* (*Xu & Beardall, 1997*; *BenMoussa-Dahmen et al., 2016*; *Hu et al., 2024*). In response to increased salinity, green microalgae generally tend to reduce the permeability of their plasma membranes by incorporating higher levels of SFAs in the phospholipid bilayer (*Mansour, Salama & Allam, 2015*). This reduction in membrane permeability is a crucial osmoregulatory mechanism that restricts the influx of Na$^+$ ions into the cell and the leakage of osmolyte out of the cell (*Guo, Liu & Barkla, 2019*).

Interestingly, the PUFA content of the biomass grown at a salinity of 1.0 M amounted to 47.4%, with a particularly high amount of alpha-linolenic acid (36.9%). The content of this omega-3 fatty acid is much higher than what reported by *Perdana et al. (2021)* for other species of *Dunaliella*. This essential (omega-3) fatty acid is of relevance to the nutraceutical sector because dietary intake of alpha-linolenic acid is associated with various health benefits, such as reduced risk of cardiovascular events, diabetes and inflammatory diseases (*Pandohee, 2022*).

**Table 4** Effect of salinity on the fatty acid profile (mass % of total fatty acids) of the *Dunaliella* isolate SCH18.

| Fatty acids methyl esters (mass %) | NaCl concentration | | | | |
|---|---|---|---|---|---|
| | 0.5 M | 1.0 M | 2.0 M | 3.0 M | 4.0 M |
| Myristic acid (C14:0) | 0.5 | – | 0.5 | 0.7 | 0.8 |
| Palmitic acid (C16:0) | 17.0 | 18.0 | 19.7 | 20.8 | 22.9 |
| Palmitoleic acid (C16:1) | 3.3 | 2.8 | 2.7 | 2.9 | 2.5 |
| Stearic acid (C18:0) | 1 | – | 0.9 | 1.5 | 1 |
| Cis-9-oleic acid (C18:1) | 2.6 | 2.3 | 3.5 | 3.6 | 4.2 |
| Linoleic acid (C18:2) | 5.2 | 7.2 | 10.8 | 10.1 | 9.3 |
| γ-linolenic acid (C18:3n-6) | 2.6 | 3.3 | 1.8 | 1.6 | – |
| α-linolenic acid (C18:3n-3) | 26.7 | 36.9 | 24.5 | 27.3 | 28.1 |
| Behenic acid (C22:0) | – | – | – | – | 0.4 |
| Others | 41.1 | 29.5 | 35.6 | 31.5 | 30.8 |
| Total saturated fatty acid (SFA) | 18.5 | 18 | 21.1 | 23 | 25.1 |
| Total monounsaturated fatty acid (MUFA) | 5.9 | 5.1 | 6.2 | 6.5 | 6.7 |
| Total polyunsaturated fatty acid (PUFA) | 34.5 | 47.4 | 37.1 | 39 | 37.4 |

## Effect of different salinities on biodiesel quality properties

Transesterification is the most widely used method for converting lipids to FAMEs, the principal constituents of biodiesel. The fuel properties of biodiesel depend on the fatty acid composition of the oil used for its production. To be used as an automotive fuel, biodiesel is generally required to be in conformity with specifications established by the American Society for Testing Materials (ASTM D-6751) and European Committee for Standardization (EN-14214). The fuel quality of biodiesel is assessed by determining key fuel properties such as oxidation stability (OS), kinematic viscosity (KV), cetane number (CN), iodine value (IV), cold filter plugging point (CFPP), higher heating value (HHV), and cloud point (CP) among others. In this study, empirical formulas were used for the assessment of biodiesel fuel properties.

As shown in Table 5, the ADU, which is an indicator of the number of unsaturated fatty acids, ranged from 1.042 to 1.401. Biomass cultured at salinity of 1.0 M produced biodiesel with the highest degree of unsaturation, mainly due to the high PUFA content. The ADU was decreased with salt stress. The range of ADU is not specified in both ASTM D-6751 and EN-14214 standards but an increase in ADU of a given biodiesel generally leads to low cetane numbers and poor oxidative stability. For other fuel properties such as KV, ρ, CN, IV, CFPP and FP, they were found to be in accordance with the biodiesel standards ASTM D-6751 and EN-14214, indicating that the salt stress would not adversely affect the biodiesel properties.

Regarding oxidation stability, the biodiesel had an induction period ranging from 5.07 to 6.01 h. These values, which represent the allowable storage time for biodiesel, were slightly lower than the limit stipulated in the EN14214 standard. One possible cause for this low stability value is the high content of linolenic acids, which ranged from 26.3% to

**Table 5 Biodiesel fuel properties of the *Dunaliella* isolate SCH18 under different salinities.**

| Biodiesel properties | Units | NaCl concentration | | | | | Biodiesel specifications | |
|---|---|---|---|---|---|---|---|---|
| | | 0.5 M | 1.0 M | 2.0 M | 3.0 M | 4.0 M | ASTM 6751 | EN14214 |
| ADU | – | 1.04 | 1.40 | 1.06 | 1.13 | 1.09 | NS | NS |
| KV | $mm^2\ s^{-1}$ | 4.55 | 4.32 | 4.53 | 4.491 | 4.515 | 1.9–6.0 | 3.5–5.5 |
| ρ | $kg\ L^{-1}$ | 0.87 | 0.88 | 0.87 | 0.87 | 0.87 | 0.85–0.90 | NS |
| CN | – | 55.93 | 53.53 | 55.76 | 55.31 | 55.56 | ≥47 | ≥51 |
| IV | $gI_2\ 100\ g^{-1}$ | 90.21 | 116.90 | 92.06 | 97.04 | 94.22 | NS | ≤120 |
| FP | °C | 188.09 | 184.35 | 187.36 | 185.42 | 186.67 | 130 minimum | 101 minimum |
| HHV | $MJ\ kg^{-1}$ | 40.37 | 41.00 | 40.41 | 40.53 | 40.46 | NS | NS |
| CP | °C | 6.08 | 1.24 | 5.74 | 4.84 | 5.35 | NS | NS |
| CFPP | °C | −9.56 | −10.82 | −8.87 | −4.31 | −5.82 | NS | ≤ 5 ≤ −20 |
| OS | hours | 6.01 | 5.07 | 5.76 | 5.61 | 5.74 | >3 | >6 |
| Linolenic acids | wt.% | 29.30 | 40.20 | 26.30 | 28.90 | 28.10 | NS | ≤12 |
| PUFA ≥ 4 db | wt.% | 0 | 0 | 0 | 0 | 0 | NS | ≤1 |

Note:
NS, not specified; db, double bond.

40.2% (as shown in Table 5). Typically, these PUFAs contain three double bonds, which are highly reactive with oxygen and therefore more prone to auto-oxidation. For this reason, the content of linolenic acids is restricted to 12% in the EN 14214 standard. *Knothe (2007)* have suggested several approaches to improve the stability of biodiesel, and one of these is the addition of antioxidant additives. Another alternative could be to selectively decrease the content of linolenic acids by employing silver ion chromatography (*Nagappan & Kumar Verma, 2018*). Using this technique, the latter were able to decrease the content of alpha-linolenic acid in *Desmodesmus* sp. MCC34 by 92%, leaving the residual oil with a higher amount of saturated and monounsaturated fatty acids.

Another important fuel characteristic that is worth mentioning is the higher heating value (HHV), or heat of combustion. Although this parameter is not specified in the international biodiesel specifications, the European standard EN 14213 for biodiesel has set a minimum HHV of 35 MJ kg$^{-1}$ (*Mata et al., 2013*). The reported HHVs for various traditional biodiesel and diesel fuels are around 39–41 and 46 MJ kg$^{-1}$, respectively (*Demirbas, 2009*). Therefore, the obtained HHV of 40.3 to 41.0 MJ kg$^{-1}$ was in accordance with the limit set by EN 14213 and comparable to the value of 40.2 MJ kg$^{-1}$ reported for the biodiesel produced from *D. tertiolecta* (*Tizvir et al., 2022*).

## CONCLUSIONS

To the best of our knowledge, this is the first strain of *Dunaliella* from a Mauritian saltern that has been successfully isolated and characterized. The isolated strain was identified as *D. tertiolecta* SCH18 based on morphological characteristics and molecular analyses of the ITS region and the 18S rDNA gene. This identification was supported by the physiological characteristics of the organism, namely optimal growth at a salinity level of 1.0 M NaCl and inability to hyperaccumulate carotenoids. The results showed that salt concentration

has a significant effect on the biochemical constituents of the isolate. Under salinity stress, the protein, lipid and carbohydrate content of the isolate decreased from $30.35 \pm 0.18\%$ to $18.48 \pm 2.16\%$, $17.81 \pm 2.40\%$ to $12.67 \pm 0.48\%$ and $53.14 \pm 2.89\%$ to $18.67 \pm 2.51\%$, respectively. The main fatty acids along the different salinities were palmitic, stearic, palmitoleic, oleic, linoleic, and linolenic acids, which are suitable for producing biodiesel. The fuel properties were in accordance with the specifications of the ASTM 6751 standard. The results obtained in this study suggest that the *D. tertiolecta* SCH18 strain could be of interest for the production of biodiesel. However, further research should be carried out to optimize cultivation conditions to achieve a higher lipid content in order to make the production of microalgae-based biodiesel economically viable.

### Funding

This work was supported by the University of Mauritius, grant number Q0191. This study also received funding from the Researchers Supporting Project at King Saud University, Riyadh, Saudi Arabia (Fund no. RSP2024R487). The funders had no role in study design, data collection and analysis, decision to publish, or preparation of the manuscript.

### Grant Disclosures

The following grant information was disclosed by the authors:
University of Mauritius: Q0191.
King Saud University: RSP2024R487.

### Competing Interests

The authors declare that they have no competing interests.

### Author Contributions

- Kamlesh Ramdhony conceived and designed the experiments, performed the experiments, analyzed the data, prepared figures and/or tables, authored or reviewed drafts of the article, and approved the final draft.
- Daneshwar Puchooa conceived and designed the experiments, prepared figures and/or tables, authored or reviewed drafts of the article, and approved the final draft.
- Turki Kh. Faraj conceived and designed the experiments, prepared figures and/or tables, authored or reviewed drafts of the article, and approved the final draft.
- Abdulwahed Fahad Alrefaei analyzed the data, prepared figures and/or tables, authored or reviewed drafts of the article, and approved the final draft.
- JunFu Li analyzed the data, prepared figures and/or tables, authored or reviewed drafts of the article, and approved the final draft.
- Rajesh Jeewon conceived and designed the experiments, performed the experiments, analyzed the data, prepared figures and/or tables, authored or reviewed drafts of the article, and approved the final draft.

## Data Availability

The DNA sequence data is publicly available in GenBank with accession number OR158048.

## Supplemental Information

Supplemental information for this article can be found online at http://dx.doi.org/10.7717/peerj.18325#supplemental-information.

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
