# Peer review of "A polyphasic approach in the identification and biochemical characterization of Dunaliella tertiolecta with biodiesel potential from a saltern in Mauritius"

_PeerJ, doi:10.7717/peerj.18325_

## Round 0.1 · original submission · Major Revisions

Thank you for submitting your manuscript titled "Physiological, Morphological, and DNA Sequence-Based Analysis Revealed Potential Strain for Biodiesel Production." We appreciate your efforts in exploring this important topic. After a detailed review, I would like to provide some feedback and suggestions for improving the manuscript.

The reviewers have acknowledged the novelty of your work, particularly its potential to lay the groundwork for future studies on Dunaliella tertiolecta in underexplored habitats. However, several key areas require significant revision to enhance the scientific rigor and clarity of your study. The following points summarize the major concerns raised by the reviewers:

The current title of your manuscript is somewhat misleading and does not fully reflect the scope of the study. Reviewer #3 pointed out that while the title suggests a focus on physiological data, your manuscript lacks estimation of photosynthetic parameters. Additionally, the focus on DNA sequence-based analysis seems more aligned with phylogenetic analysis rather than a broader sequence-based approach. I recommend revising the title to better align with the key findings and objectives of your research, ensuring clarity and consistency for readers.

Reviewer #1 raised concerns about the drying method used for algal biomass at 60˚C, particularly its potential to denature proteins and pigments. It is crucial to provide more details on the duration of drying and to discuss whether this method is suitable for the extraction procedures used. If this method was chosen deliberately, please provide references that justify its use, and consider discussing its impact on the yields of bio-compounds.

Both reviewers emphasized the need to incorporate and compare your findings with recent and relevant literature. For example, Reviewer #3 noted the absence of comparisons with other studies on Dunaliella tertiolecta, which could help situate your findings within the broader context of existing research. Specific references were suggested, such as those by Park et al. (2015), Singh et al. (2019), and Kim et al. (2024). Integrating these references will not only strengthen the discussion but also highlight the relevance and novelty of your work.

Reviewer #3 suggested that the data presentation could be improved by including a table to summarize the findings related to protein, pigments, carbohydrates, and lipids. Additionally, incorporating Nile red-based fluorescence microscopy or better photographic evidence to show the behavior of cell shape and size under salt exposure could significantly enhance the visual clarity and impact of your findings.

Reviewer #3 Also expressed concerns regarding the categorization of the strain as non-carotenogenic while reporting maximum carotenoid levels at 1M NaCl. This appears contradictory to the existing literature on the increase of carotenoids with salt stress. It is important to provide a clear and well-justified explanation for this finding, or to reconsider how these results are presented.
Given these significant issues, I strongly encourage you to revise your manuscript accordingly. The incorporation of recent literature, improvement in data presentation, and clarification of your methodological approach will greatly enhance the quality and impact of your study.

Reviewer 1 ·

Basic reporting

No comment

Experimental design

No comment

Validity of the findings

No comment

Additional comments

The manuscript is well prepared and structured including citations of pioneering works related to Dunaliella species. The work is novel and can provide groundwork for future studies based on D. tertiolecta, especially in habitats that have not been previously studied.

However, some suggestions that can be considered before publication:

Major comments:
1. In lines 208-211, the authors have mentioned drying algal biomass at 60˚C before extracting the carbohydrates, proteins, lipids and pigments. The authors have not mentioned the duration undertaken for drying algal biomass at such high temperatures, as most of the pigments and proteins get denatured at such high temperatures. Is this a suitable method for using algal biomass for extraction procedures? If so, then adding a reference is advised. Also, have the authors considered the effect of this method on the yields of the aforementioned bio-compounds?
Minor comments-
1. Lines 48-57, discuss about the economic importance of algal biofuels. The authors are advised to add suitable citations for this data. Apart from this, references can be based on techno-economic analysis. For example-
Valdovinos-García, E. M., Bravo-Sánchez, M. G., Olán-Acosta, M. de los Á., Barajas-Fernández, J., Guzmán-López, A., & Petriz-Prieto, M. A. (2022). Technoeconomic Evaluation of Microalgae Oil Production: Effect of Cell Disruption Method. Fermentation 2022, Vol. 8, Page 301, 8(7), 301. https://doi.org/10.3390/FERMENTATION8070301

Llamas, B., Suárez-Rodríguez, M. C., González-López, C. v., Mora, P., & Acién, F. G. (2021). Techno-economic analysis of microalgae related processes for CO2 bio-fixation. Algal Research, 57, 102339. https://doi.org/10.1016/J.ALGAL.2021.102339

2. It is suggested to add suitable references (preferably from 2015 onwards) in the second paragraph of the “Introduction” section.
3. In lines 435-438, adding a reference is advised.
4. In lines 569-570- the authors write, “Another alternative could be the mixing of the microalgal oil produced with more saturated oil such as palm and coconut oil.” Authors are suggested to substantiate these lines through some citation or any industrial work if available rather than providing general statements.

Reviewer 2 ·

Basic reporting

Clear and unambiguous, professional English was used throughout. Literature references, sufficient field background/context was provided. Professional article structure, figures, tables, raw data were shared.

Experimental design

1. ITS can only identify microalgae at genus level, not species level.
2. A low lipid content (17.81± 2.4%) was very not suitable for biodiesel production.
3. This study was very simple.

Validity of the findings

1. A low lipid content (17.81± 2.4%) was very not suitable for biodiesel production.
2. This study was very simple.

Additional comments

Physiological, morphological and DNA sequence-based analysis reveal Dunaliella teriolecta SCH18 as a halotolerant microalga from a solar saltern - a potential strain for biodiesel production
About this paper:
Biodiesel is a promising bioenergy source. In this study, a microalgal strain SCH18 was isolated from a solar saltern located in Mauritius. The authors identified this strain and explore its potential in producing biodiesel. However, this paper has a very significant drawback.
1. ITS can only identify microalgae at genus level, not species level.
2. A low lipid content (17.81± 2.4%) was very not suitable for biodiesel production.
3. This study was very simple.
Therefore, I suggested that this paper was not suitable for publication in PeerJ.

·

Basic reporting

The article should include sufficient introduction and background to demonstrate how the work fits into the broader field of knowledge. Relevant prior literature should be appropriately referenced. References are not up to date.

Experimental design

Methods are not described with sufficient details and physiological parameters have not been studied.
There is a knowledge gap in the current study and authors have not done an elaborate literature survey.

Validity of the findings

Impact and novelty not assessed

Additional comments

The paper entitled “physiological, morphological and DNA sequence-based analysis revealed potential strain for biodiesel production”. The title of the manuscript is misleading and does not accurately reflect the scope and content of the study presented.
• While the title suggests a focus on physiological data, the actual work described in the manuscript lacks the estimation for photosynthetic parameters.
• The workers have described it as DNA sequence-based analysis, however the methodology and focus of the study appears to align more closely with the phylogenetic analysis.
I recommend revising the title to better align with the key findings and objectives of the research to ensure clarity and consistency for the reader.
There are many reports of Dunaliella tertiolecta sp. in the literature, it may be the first report on the Mauritius strain but overall, the work lacks novelty as this ms is a mere amalgamation of previous work done on D. tertiolecta. Specific studies with relation of overproduction strategies of lipids; carbohydrate contents are properly studied in the cultivation strategy.
The manuscript lacks the incorporation of recent and relevant literature on the study topic and authors have not compared the data from the other workers. For e.g.,:
• Park, S., Kim, M., Lee, S. G., Lee, Y., Choi, H. K., & Jin, E. (2015). Contrasting photoadaptive strategies of two morphologically distinct Dunaliella species under various salinities. Journal of Applied Phycology, 27, 1053-1062
• Singh, P., Khadim, R., Singh, A. K., Singh, U., Maurya, P., Tiwari, A., & Asthana, R. K. (2019). Biochemical and physiological characterization of a halotolerant Dunaliella salina isolated from hypersaline Sambhar Lake, India. Journal of phycology, 55(1), 60-73.
• Kim, M., Kim, J., Lee, S., Khanh, N., Li, Z., Polle, J. E., & Jin, E. (2024). Deciphering the β‐carotene hyperaccumulation in Dunaliella by the comprehensive analysis of Dunaliella salina and Dunaliella tertiolecta under high light conditions. Plant, Cell & Environment, 47(1), 213-229.

I am surprised that, workers kept it in a non-carotenogenic strain category but carotenoids are found maximum at 1M NaCl which lacks proper justification and is in contrast to the increase in carotenoids with the increase in salt stress as reported by other workers.

Even a table may be sufficient to accommodate protein, pigments, carbohydrate and lipids.
For me, this is a preliminary data and cannot be considered for publication in a good journal without major revisions. Workers, are suggested to include Nile red based fluorescence microscopy or better photographs to show the behaviour of cell shape and size under salt exposure.

---

## Round 0.2 · accepted · Accept

Dear Authors,

Thank you for submitting the revised version of your manuscript. I have carefully reviewed the changes, and I am pleased to confirm that you have successfully addressed all of the reviewers' comments.

As the previous reviewers were not available for this round of revisions, I have personally assessed the revised manuscript and am satisfied with the improvements made. Based on my review, I believe this version is now ready for publication.

Thank you for your efforts, and I look forward to seeing your work published.

Best regards,